# Unveiling *Nilaparvata lugens* Stål Genes Defining Compatible and Incompatible Interactions with Rice through Transcriptome Analysis and Gene Silencing

Priyadarshini Rout, Nihal Ravindranath [ID], Dinkar Gaikwad [ID] and Satyabrata Nanda *[ID]

MS Swaminathan School of Agriculture, Centurion University of Technology and Management, Paralakhemundi 761211, Odisha, India; priyadarshini.rout@cutm.ac.in (P.R.); nihal.r@cutm.ac.in (N.R.); gaikwad@cutm.ac.in (D.G.)
* Correspondence: satyabrata.nanda@cutm.ac.in

**Abstract:** The brown planthopper (*Nilaparvata lugens* Stål, BPH) is a major pest of rice (*Oryza sativa* L.), causing severe crop loss. Multiple biotypes and emerging populations of BPH pose a bigger challenge for the infestations control. Although several studies have been conducted to understand the molecular mechanisms of rice–BPH interactions, there are few studies dedicated to the Indian sub-continent BPH biotype (biotype 4). Here, we analyzed the transcriptomic, physiological, and gene-silencing responses of the BPH biotype 4 during the compatible (fed on susceptible Taichung Native 1, TN1 rice) and incompatible (fed on resistant PTB33 rice) rice–BPH interactions. In the incompatible interaction, a significant reduction in the honeydew production and negative weight gain were observed in the BPH. Similarly, the trehalose and glucose contents were found to be significantly high and low, respectively, during the incompatible rice–BPH interaction. The comparative BPH transcriptome analysis identified 1875 differentially expressive genes (DEGs) between the compatible and incompatible interactions from which many were annotated to be involved in vital BPH physiological processes, including cuticle development, sugar metabolism, detoxification, molting, and xenobiotics metabolism. The RNA interference-mediated independent silencing of three selected genes, including *NlCP1*, *NlCYP320a1*, and *NlTret1*, revealed that these genes are important for BPH physiology and survival. Moreover, the results of this study provide valuable insights into the rice–BPH interactions involving the BPH biotype 4.

**Keywords:** rice; brown planthopper; transcriptome; compatible and incompatible interactions; RNAi

## 1. Introduction

Rice (*Oryza sativa* L.) is the most widely accepted staple food in the world. Rice production must be accelerated to match the demand of the ever-increasing world population. However, rice production is challenged by several insect pest attacks. *Nilaparvata lugens* Stål, commonly known as brown planthopper (BPH), is one dreadful pest of rice that causes severe damage [1]. This monophagous insect feeds on the rice phloem sap, causing direct damage to rice plants. In addition, BPH acts as a vector for two rice viruses, including rice grassy and rice-ragged stunt viruses, causing indirect damage [2]. To control BPH infestations, the use of insecticides is the most practiced method worldwide. However, the over-exploitation of these synthetic insecticides brings several adverse effects, including insecticide resistance, elimination of natural enemies, insect resurgence, and environmental hazards. On the other hand, several rice molecular breeding programs have resulted in varieties showing improved resistance to BPH, which is considered to be a preferred alternative strategy for BPH control [3]. Specific genes providing resistance (*Bph/bph* genes) to BPH have been identified in rice. For instance, the first two *Bph* genes, *Bph1* and *Bph2*, were identified in 1970 from Mudgo and ASD7 rice varieties [4]. To date, 45 *Bph* genes have been identified in rice varieties and their wild relatives [5].

BPH has co-evolved with rice and thus has developed several strategies to adapt or outrun the rice defenses [6]. For instance, the resistance response of the rice varieties containing *Bph* genes was broken down by the emergence of new virulent BPH populations [7,8]. Although the exact mechanism underlying this is largely unknown, it is highly probable that the BPH virulence genes or effectors play a major role in it. Further, the role of epigenetics in contributing towards the pest phenotypic or population variance can be an additional factor for this [9]. The effect of climate change and other environmental factors can cause heritable epigenetic modifications in insects, enabling a rapid evolution [10]. In hemipteran insects, including BPH, DNA methylation has been reported to be a crucial epigenetic modification in regulating several physiological processes [11,12]. A recent study has reported that the heritable epigenetic modifications, particularly the stress-induced DNA methylation in BPH, contributed to overall stress resilience and rapid adaptions [13]. Additionally, the role of BPH metabolites, host-specific responses, and BPH endosymbionts aid in the process of rice resistance breakdown [14–16]. The BPH infestation on different rice varieties can influence its feeding and overall fitness, including growth, metabolism, and gene expression [1,17,18]. For example, the amount of honeydew produced, which is often considered as an indicator of BPH fitness, during the BPH feeding on a resistant rice variety is significantly less than the BPH feeding on a susceptible rice variety [18,19]. Likewise, changing the host from susceptible rice (Taichung Native 1, TN1) to resistant rice (B5 rice with *Bph14* and *15*) induced the expression of genes related to trehalose synthesis [20]. Overall, BPH feeding on a resistant rice variety (incompatible interaction) tends to have an elevated amount of trehalose as compared to BPH feeding on a susceptible rice variety (compatible interaction). In contrast, the glucose content is elevated in BPH during a compatible interaction as compared to an incompatible interaction [18,20]. Similarly, the type of interaction, i.e., compatible or incompatible, can reprogram the BPH transcriptome. Moreover, comparative transcriptomic studies have been carried out in BPH to find out the key regulating genes, microRNAs, and potential effectors [18,20–24]. All these studies have reported the differential BPH transcriptome while feeding on a resistant and a susceptible rice variety and have identified several differentially expressed genes (DEGs). However, only a few of these studies focused on the whole-body transcriptome, while most analyzed the salivary gland transcriptomes. Moreover, all these studies have been done on the BPH biotypes 1 and 2, which are the abundant types in China and other East and Southeast Asia areas. Conversely, very limited information is available on the transcriptome levels of the Indian subcontinent BPH biotype (biotype 4). As it has already been established that different BPH biotypes have varied levels of virulence and host preferences, it will be hugely beneficial to identify and analyze the transcriptomes of the BPH biotype 4 during its infestation on a resistant and susceptible rice variety.

In this study, the BPH biotype 4 originally reared on TN1 rice was fed on two contrasting varieties of rice, TN1 (susceptible, has no *Bph* gene) and PTB33 (resistant, has *BPH2*, *BPH17-ptb*, and *BPH32* genes). The overall BPH fitness was analyzed by measuring the honeydew amounts and BPH weight gains. Additionally, two major sugars, glucose and trehalose, were quantified for the BPH fed on two separate rice varieties. The whole-body transcriptomes of the BPH fed on respective rice varieties were compared and analyzed to find out DEGs and key regulating genes. Finally, the RNA interference (RNAi)-based survival assays were carried out to find out the silencing effects of some selected genes. Overall, the results of this study will help in understanding the transcriptome dynamics of the Indian subcontinent BPHs during compatible and incompatible interactions.

## 2. Materials and Methods

### 2.1. Plant and Insect Materials

Two *indica* rice varieties, including PTB33 (resistant to BPH) and TN1 (susceptible to BPH), and BPH colonies were originally collected from the Indian Council of Agricultural Research-National Rice Research Institute, Cuttack, Odisha, India. The rice varieties were grown and maintained at Centurion University of Technology and Management (CUTM),

Paralakhemundi, Odisha, India. The BPH colonies were maintained on the TN1 rice for more than 30 generations in the insectary at the CUTM campus at $28 \pm 2$ °C and $75 \pm 5\%$ relative humidity under a 14/10 h light/dark photoperiod. The PTB33 and TN1 rice seedlings (45 days old) and the newly emerged BPH females were used as the plant and insect material, respectively, in this study.

## 2.2. BPH Bioassays

The BPH bioassays were conducted as described by Wan et al. [18]. Briefly, the newly emerged females were collected, weighed, and put in individual Parafilm sachets ($3 \times 3$ cm). Subsequently, the sachets were attached to the TN1 (compatible) or PTB33 (incompatible) rice plant. The sachets were collected after 48 h, and the weights of the insect and sachet were measured. The difference in weight of the BPH before putting in the sachets and post 48 h infestation period was defined as the BPH weight gain. Similarly, the difference in weight of the empty Parafilm sachet and post 48 h infestation period excluding the BPH was defined as honeydew weight. The experiment was conducted with 10 replicates, where each replicate constituted 20 BPHs.

## 2.3. Transcriptome Sequencing and Analysis

In total, 50 BPH individuals were transferred onto TN1 and PTB33 rice separately. After 48 h of infestation, the BPH was collected, immediately frozen with liquid nitrogen, and stored at $-80$ °C. The Total RNA from the BPH was isolated using the TRIzol Reagent (Invitrogen, Carlsbad, CA, USA). The quantity and quality of the isolated RNA were estimated on a NanoDrop1000 spectrophotometer (Thermo Fisher Scientific, Waltham, MA, USA) and by running a 0.8% agarose gel electrophoresis. From the isolated RNA samples, the cDNA libraries were constructed and sequenced on an Illumina HiSeq 2000 sequencing platform at Wipro Genomics (Bengaluru, Karnataka, India). The paired-end raw sequence reads were assessed for base quality and contamination by sequencing artifacts. Trimming of adapters and poor-quality sequences was performed for paired sequence reads with Trim Galore. The trimmed sequence reads were mapped to reference with the splice-aware alignment tool STAR. The feature-specific expression counts were estimated by using the R-subread R package. The detection and removal of low-count features across samples was performed by using the NOISeq R package, and the expression counts normalization was performed by using the Trimmed Mean of M-values (TMM) method. The estimation of significant differential features was performed with the NOISeq R package, while the functional annotation for the protein-coding genes was performed with the eggNOG-mapper tool. Gene ontology summarization and treemap plot were generated with the R package rrvgo.

## 2.4. Expression Analysis via RT-qPCR

The expression analysis of the selected BPH genes was performed on a Roche Light Cycler real-time quantitative PCR (RT-qPCR) machine (Basel, Switzerland) by using the PowerTrack™ SYBR Green Master Mix (Thermo Fisher Scientific, Waltham, MA, USA). The BPH *tub* and *rps15* genes were taken as the internal reference genes (Table S1) [18]. The relative expression was calculated by following the $2^{-\Delta\Delta CT}$ method. The experiment contained three biological replicates, and each biological replicate contained three technical triplicates.

## 2.5. Analysis of Glucose and Trehalose Contents in BPH

The contents of glucose and trehalose were analyzed as described by Wan et al. [18]. In short, the BPH samples were collected and homogenized in 1 mL pre-cooled phosphate-buffered solute on (PBS 0.02 M, pH 6.0). The homogenized mixture was then centrifuged at 3000 rpm for 15 min at 4 °C. Subsequently, the supernatant was collected and equally divided into two portions. The first portion was used to measure the trehalose content by using the trehalose assay kit (Neogen, Lansing, MI, USA). Similarly, the second portion was centrifuged at 13,000 rpm for 50 min at 4 °C, and the supernatant was collected to estimate

the glucose content by using a glucose (GO) assay kit (Sigma-Aldrich, St. Louis, MO, USA). The experiment was conducted with ten replicates.

### 2.6. Double-Stranded RNA (dsRNA) Synthesis and RNAi Assays in BPH

Specific forward and reverse primers containing the $T_7$ promoter sequence were used to amplify *NlCP1 (LOC111046236)*, *NlCYP320a1 (LOC111060963)*, *NlTret1 (LOC111058179)*, and *green fluorescent protein (GFP)* fragments for dsRNA synthesis (Table S2). The resulting PCR products were purified by using a Wizard® SV Gel and PCR Clean-Up System (Promega, Madison, WI, USA). The purified PCR products were used as the template to synthesize dsRNA with the $T_7$ MEGAscript kit (Thermo Fisher Scientific, Waltham, MA, USA). A NanoDrop1000 spectrophotometer (Thermo Fisher Scientific, Waltham, MA, USA) was used to detect the dsRNA concentrations. The synthesized dsRNAs were stored at $-80\ °C$ until use.

To perform the RNAi bioassays, specific dsRNAs were injected into the BPH as previously described [18]. Briefly, 200 ng of the dsRNAs (individual target genes or *GFP*, negative control) was injected into the abdomen of the newly emerged fourth instar BPH. After the dsRNA injections, the BPH was reared on TN1 or PTB33 rice to start the bioassay. For each dsRNA treatment, the experiment was performed by including 120 fourth-instar nymphs (6 replicates, 20 individuals per replicate). Out of the six replicates, three were used for BPH survival analysis, and the other three were used for expression evaluation by RT-qPCR.

### 2.7. Statistical Analysis

The statistical analysis of the experimental data was performed by using the data processing system. To compare two samples, the Student's *t*-test was employed. Similarly, to compare three or more samples, one- or two-way analysis of variance (ANOVA) followed by Tukey's HSD post hoc test was used. The statistical significance was evaluated at $p < 0.05$.

## 3. Results

### 3.1. BPH Performance on TN1 and PTB33 Rice

The performance of the BPH fed on the TN1 and PTB33 rice varieties was evaluated to analyze the overall BPH fitness during the compatible and incompatible interactions, respectively. The amount of honeydew excreted during the compatible interaction was significantly higher ($p < 0.01$) than that during the incompatible interaction (Figures 1A and S1). Similarly, the weight gain for the BPH fed on TN1 rice was significantly higher ($p < 0.01$) than the BPH fed on PTB33 rice, which had a negative weight gain (Figure 1B).

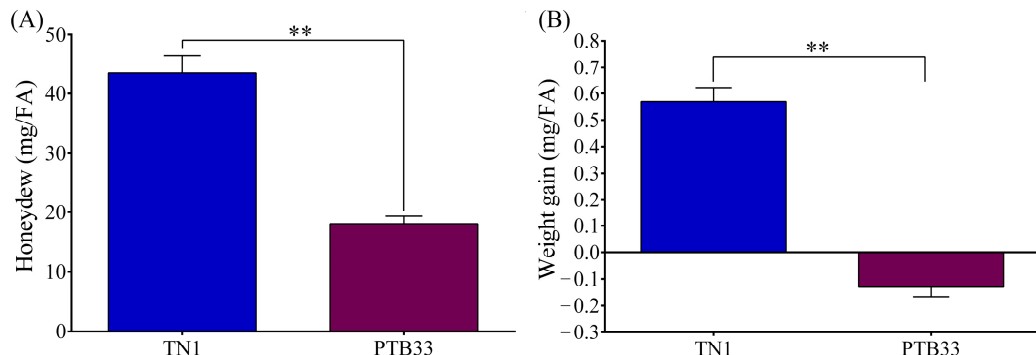

**Figure 1.** The performance evaluation of BPH fed on TN1 and PTB33 rice separately: (**A**) comparison between the honeydew amounts secreted; (**B**) comparison between the weight gains. Data are denoted as mean $\pm$ SE ($n = 200$). The double asterisks indicate a statistically significant difference between TN1 and PTB33 rice-fed BPH, respectively, at $p < 0.001$. The unit mg/FA denotes for milligram per female adults.

### 3.2. Analysis of Glucose and Trehalose Content in the BPH Infested on TN1 and PTB33 Rice

Infestation of different rice varieties can influence the sugar metabolism of BPH. To investigate this, we analyzed the glucose and trehalose contents of the BPH fed on TN1 or PTB33 rice. The results revealed that the glucose content was significantly higher ($p < 0.01$) in the BPH fed on TN1 rice as compared to the BPH fed on PTB33 rice (Figure 2A). On the contrary, the trehalose content was found to be lower ($p < 0.05$) in the TN1 rice-fed BPH than in the PTB33 rice-fed ones (Figure 2B). The glucose to trehalose ratio was also found to be significantly higher ($p < 0.01$) in the TN1 rice-fed BPH as compared to the PTB33 rice-fed BPH.

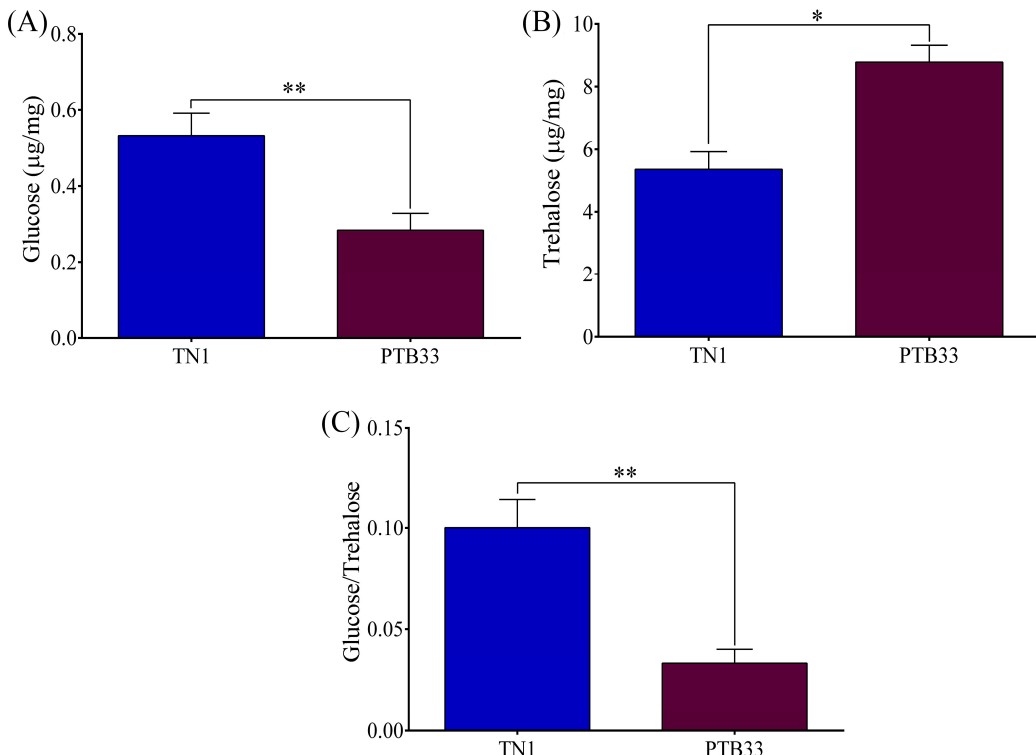

**Figure 2.** The different sugar content estimation of BPH fed on TN1 or PTB33 rice: (**A**) comparison between the glucose content; (**B**) comparison between the trehalose content; (**C**) comparison between the ratio of glucose/trehalose content. Data are denoted as mean ± SE ($n = 200$). Single asterisk indicates a statistically significant difference between TN1 and PTB33 rice-fed BPH, respectively, at $p < 0.05$, and double asterisks indicate the statistical significance at $p < 0.01$.

### 3.3. Identification and Analysis of DEGs

In our attempt to investigate the BPH transcriptomes during the compatible (fed on TN1 rice) and incompatible (fed on PTB33 rice) interactions, the respective RNA libraries were sequenced. In total, 34.18 and 34.97 million clean reads were obtained from the BPH_TN1 (BPH fed on TN1 rice) and BPH_PTB33 (BPH fed on PTB33 rice) libraries, respectively (Tables 1 and S4). From the obtained reads, about 91.5% of the reads were mapped to the BPH genome. From the mapped reads, 14,454 genes were found to be expressive and subsequently, comparatively analyzed in BPH_TN1 and BPH_PTB33 groups for the identification of the DEGs. In total, 1875 genes were found to show significant differential expressions (765 upregulated, 1110 down regulated) between BPH_TN1 and BPH_PTB33 ($P_{adj} < 0.05$) (Table S3, Figure 3). Out of these identified DEGs, several genes are predicted to be involved in crucial BPH physiological processes, such as cuticle formation (*LOC111046236, LOC111046246, LOC111057579, LOC111060728,* and *LOC111059889*), metamorphosis (*LOC111063850, LOC111058709, LOC111060144,* and *LOC111058933*), oxidoreductase process (*LOC111045053, LOC111045250, LOC111046371,* and *LOC111046190*),

and protein folding (*LOC111059963*, *LOC111059961*, and *LOC111054162*), were found to be significantly down-regulated (fold change $< -1$, $P_{adj} < 0.05$) in the BPH fed on PTB33 rice as compared to the BPH fed on TN1 rice. Similarly, genes involved in sugar transport (*LOC111058179*, *LOC111057792*, *LOC111051411*, and *LOC111048003*), calcium binding (*LOC111050001*, *LOC111060644*, *LOC111051627*, and *LOC111047883*), detoxification (*LOC111060963*, *LOC111060963*, and *LOC111059139*), and lipid metabolism (*LOC111058678*, *LOC111045597*, *LOC111047774*, and *LOC111045596*) were found to be up regulated in the BPH_PTB33 as compared to BPH_TN1. Additionally, several important genes displayed differential transcript accumulations in the BPH fed on either TN1 or PTB33 rice (Table S1). To validate the results of the RNAseq data, we selected 10 DEGs randomly and checked their expressions via RT-qPCR analysis. The results revealed a similar trend of expression of the DEGs as obtained in the RNAseq results (Figure 4). However, the extent of expressions was not the same as obtained in RNAseq data.

**Table 1.** The summary of RNA-sequencing result data. The raw sequences were analyzed to obtain the trimmed data by removing adapter and poor-quality sequences and were aligned to the reference transcriptome.

| Sample Name | Raw Sequence | Trimmed Sequence | % GC | Average Length | % Aligned |
|---|---|---|---|---|---|
| BPH_PTB33 | 34.2 million | 34.18 million | 35 | 141 | 92 |
| BPH_TN1 | 35.08 million | 34.97 million | 35 | 140 | 91 |

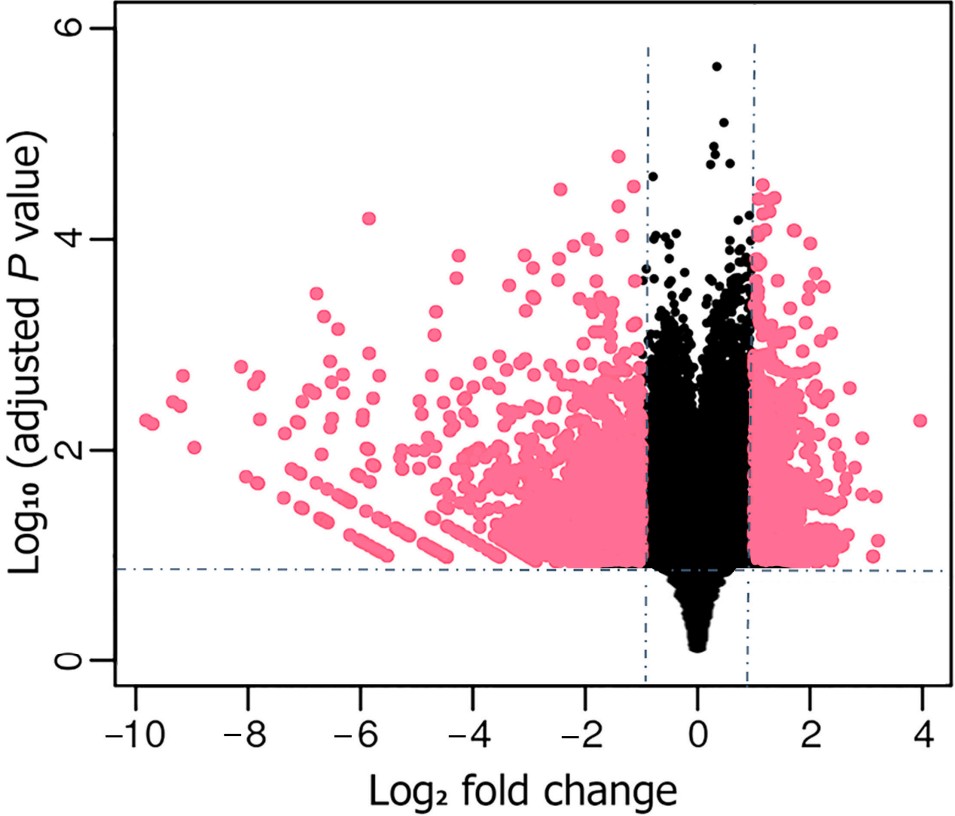

**Figure 3.** The differentially expressed genes (DEGs) were identified in BPH feeding on TN1 and BPH rice for 48 h ($n = 100$). The DEGs are shown as pink dots on the X-axis ($\log_2$ fold change). The negative value represents a downregulation, and the positive value represents an upregulation of a DEG. DEGs showing no statistical significance in their expressions are shown in black color. The $\log_2$ fold change cutoff is represented by the vertical dashed line, and the $P_{adj}$ value cutoff is represented by the horizontal dashed line.

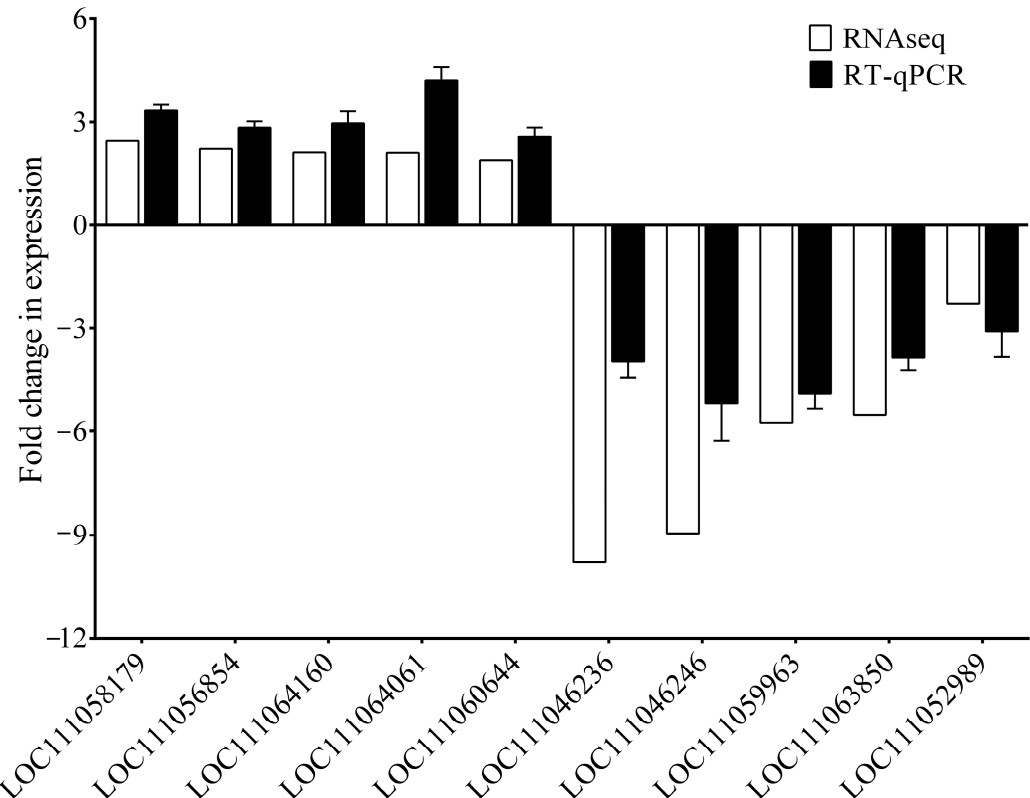

**Figure 4.** Comparison between the expression levels of the 10 selected DEGs between BPH_PTB33 and BPH_TN1 as obtained from the RNAseq data and RT-qPCR analysis (*n* = 100). The expression is shown in fold change values (Y-axis). On X-axis, the gene IDs of the selected DEGs are mentioned.

The DEGs identified in this study were subjected to the GO and KEGG enrichments to deduce the functional annotations. The GO analysis results revealed the predicted functions of the DEGs in three different categories, including biological, molecular, and cellular. The major annotated biological functions of the DEGs included metabolic processes (GO:0090304, GO:0046483, GO:0044260, GO:0044237, GO:0043170, GO:0034641, and GO:0006139), cuticle development (GO:0042335, GO:0040005, GO:0040003, and GO:0008362), protein modifications (GO:0043412 and GO:0036211), and molting (GO:0007591) (Figure 5A). Similarly, the most annotated molecular functions of the DEGs were found to be cyclic compound binding (GO:1931363 and GO:0097159), structural constituents of cuticle (GO:0042302, GO:0008010, and GO:0005214), protein and nucleic acid binding (GO:0005515, GO:0005488, and GO:0003676), and enzymatic activities (GO:0034338, GO:0017171, and GO:0008236). Additionally, the GO results suggested that genes involved in one of the crucial physiological processes, i.e., cuticle development, were significantly downregulated in BPH fed on PTB33 rice. On the other hand, genes involved in processes, such as carboxylesterase activity, are significantly upregulated in the BPH fed on PTB33 rice as compared to the BPH fed on TN1 rice. The KEGG enrichment results revealed that the identified DEGs could be involved in vital BPH pathways (Figure 5B). For instance, two important pathways were found enriched in the KEGG analysis, including xenobiotics degradation and metabolism and development and regeneration. The transcriptome and GO analysis results confirmed that several DEGs have been annotated with the function carboxylesterase activity, which is usually associated with the xenobiotic metabolism in BPH. Similarly, several DEGs were annotated to be involved in cuticle development, which is further supported by the KEGG enrichment results.

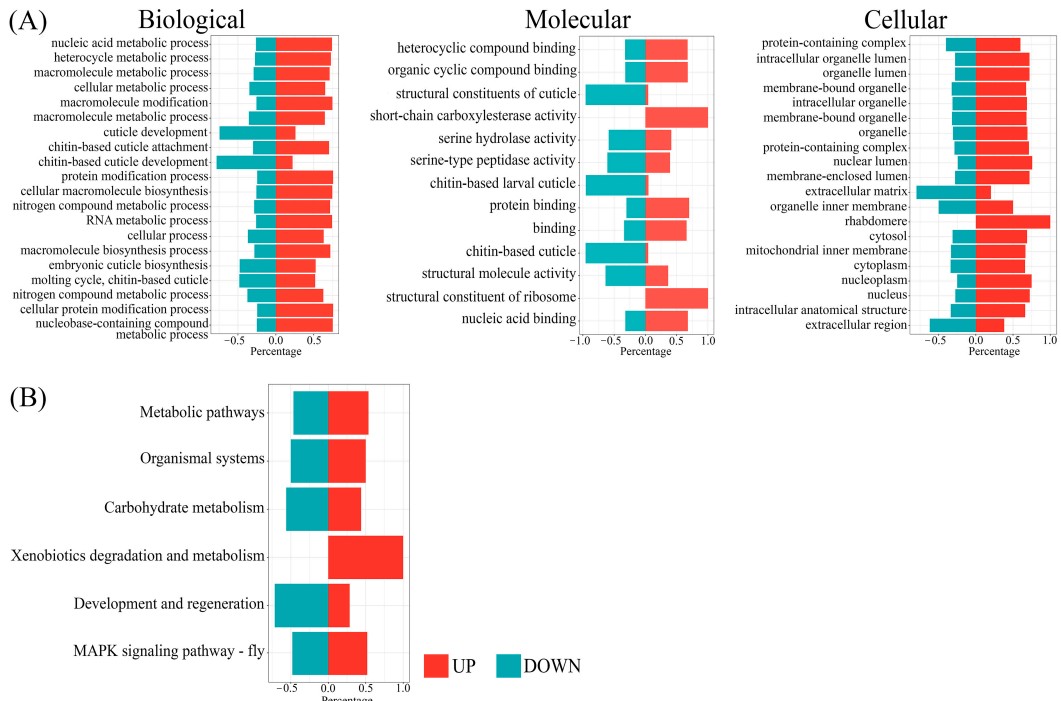

**Figure 5.** GO analysis (**A**) and KEGG pathway enrichments (**B**) of the identified DEGs in BPH. The GO terms or KEGG pathway information is mentioned at the left of each figure. The red portion represents the upregulated DEGs, while the teal portion represents the downregulated DEGs.

### 3.4. Silencing of Selected DEGs and Effect Analysis

After the GO and KEGG functional annotations of the DEGs, we selected three BPH genes, including *LOC111046236* (encoding a cuticle protein, *NlCP1*), *LOC111060963* (encoding a cytochrome p450 protein, *NlCYP320a1*), and *LOC111058179* (encoding a trehalose transporter, *NlTret1*). Out of these selected genes, *NlCYP320a1* and *NlTret1* exhibited significantly upregulated expression in the BPH fed on PTB33 rice, whereas *NlCP1* displayed significant downregulation in the BPH fed on PTB33 rice as compared to the BPH fed on TN1 rice. These genes were selected to validate their functions and to access their silencing effects in BPH via RNAi. The RT-qPCR results confirmed that all three gene expressions were significantly suppressed (Figure 6). On the other hand, the survival assay results confirmed that all the selected genes are essential for the survival of BPH and their silencing caused BPH mortality (Figure 7). The injection of ds*NlCP1* caused significant mortality in the BPH fed on PTB33 rice after 3 days post injection (DPI), while significant mortality was seen in the BPH reared on TN1 rice after 5 DPI (Figure 6). Similarly, administration of ds*NlCYP320a1* caused significant mortality in the PTB33-fed BPH on 2 DPI, whereas in the TN1-fed BPH, significant mortality was observed on 4 DPI (Figure 6). Interestingly, in the ds*NlTret1* injected BPH, significant mortality was seen after 2 DPI for the PTB33-fed BPH, while on 10 DPI any significant mortality was to be seen in the BPH fed on TN1 rice (Figure 6). Silencing of these genes induced mortality in the BPH fed on either TN1 or PTB33 rice. All the BPH injected with the control ds*GFP* could develop normally and have a significantly higher survival rate than the dsRNA-injected ones.

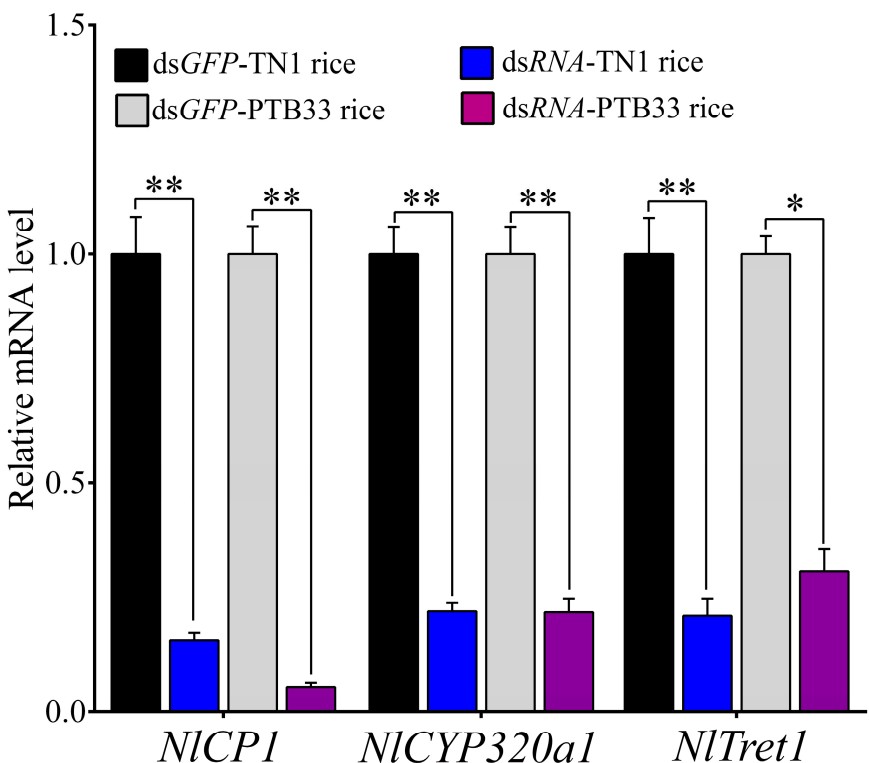

**Figure 6.** The effects of dsRNA (ds*NlCP1*, ds*NlCYP320a1*, and ds*NlTret1*) on the corresponding gene expressions. Specific dsRNAs (individual target genes or GFP, negative control, 200 ng) were injected into the abdomen of the newly emerged fourth instar BPH. The gene expressions were analyzed at 48 h post-injection. Data are denoted as mean $\pm$ SE (*n* = 120). Single asterisk indicates a statistically significant difference between the control and test condition at *p* < 0.05, and double asterisks indicate the statistical significance at *p* < 0.01, respectively.

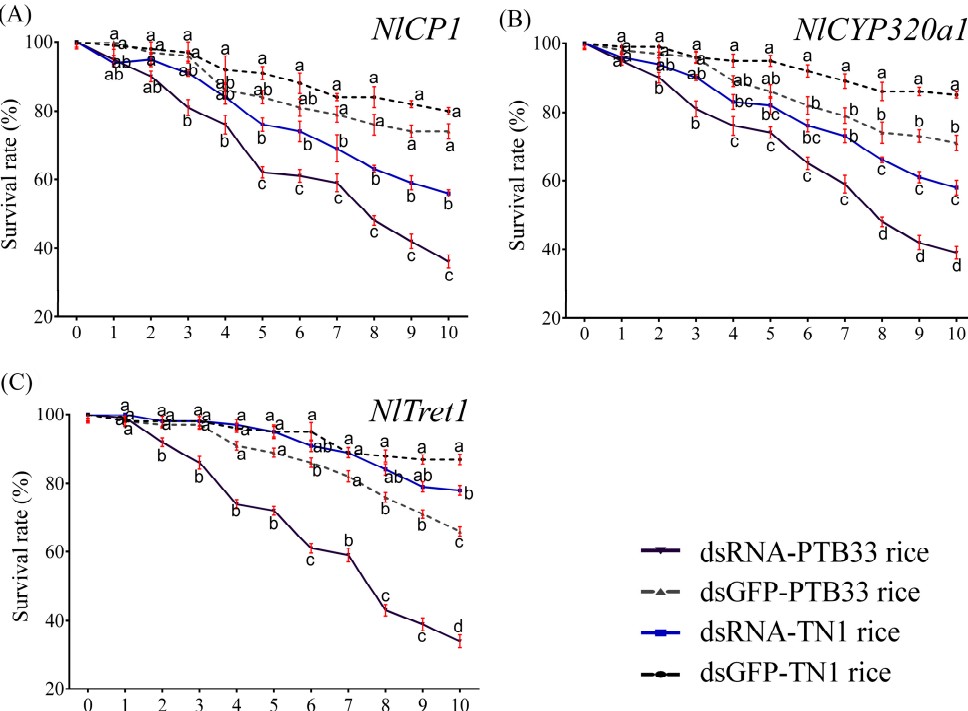

**Figure 7.** The BPH survival assays on TN1 and PTB33 rice. The BPH survival rates after the treatments of (**A**) ds*NlCP1*, (**B**) ds*NlCYP3230a1*, (**C**) ds*NlTret1*. The injection of ds*GFP* served as the control for

the RNAi assay. Specific dsRNAs (individual target genes or GFP, negative control, 200 ng) were injected into the abdomen of the newly emerged fourth instar BPH. Data are denoted as mean $\pm$ SE ($n$ = 120). Statistical significance (alphabets a–d) was analyzed by two-way ANOVA followed by Tukey's HSD post-hoc test at $p < 0.05$.

## 4. Discussion

The rice–BPH interactions are complex. The molecular mechanisms underlying these complex interactions are being explored worldwide. Several recent studies have suggested different mechanisms of rice resistance and BPH adaptation to rice [3,25–29]. In this study, the phenotypic and transcriptomic responses of the Indian sub-continent biotype 4 BPH fed on two contrasting rice varieties, TN1 (susceptible) and PTB33 (resistant), have been investigated. The amount of honeydew excreted is considered to be an indicator of BPH fitness on rice as the honeydew amount is directly proportional to the amount of phloem sap ingested [18]. Here, the honeydew produced during the incompatible interaction (BPH feeding on PTB33 rice) was significantly lower than the honeydew produced during the compatible interaction (BPH feeding on TN1 rice). The BPH while feeding on resistant rice varieties spends more time in probing than feeding [19]. Similar feeding patterns have been reported in aphids, where the aphids feed more and probe less in a compatible interaction and vice versa [30]. Thus, feeding the BPH on a resistant rice variety challenged the BPH fitness as witnessed by the reduced honeydew production and negative weight gain (Figure 1).

Ingestion of phloem sap is a key process in modulating the BPH sugar metabolism. Glucose and trehalose are the two major carbohydrates in the BPH sugars metabolism. In our results, we found that the glucose levels in BPH were increased during the compatible BPH–rice interaction. On the contrary, the trehalose levels were up in the BPH during the incompatible interaction. Trehalose is considered to be the main blood sugar in BPH and acts as the main energy source for BPH hopping [17,31]. Furthermore, feeding on resistant rice induced the upregulated expression of the trehalose 6-phosphate synthase gene that synthesizes trehalose from glucose in BPH [17]. Moreover, our results are corroborated by several previous studies that reported increased trehalose content in BPH while feeding on a resistant rice [17,20,32]. A reason for the elevated trehalose levels in BPH while feeding on resistant rice could be because it will supply the energy to be utilized in the extra hopping for prolonged probing [17,33,34]. Another reason could be the supplement of glucose by the hydrolysis of trehalose to meet the energy requirements for other BPH metabolisms [35]. On the other hand, the higher BPH glucose levels during the compatible interactions infer that the BPH can steadily ingest the phloem sap (made up of glucose and fructose) with significantly less probing [36]. Thus, the BPH feeding on the susceptible TN1 rice had a higher glucose level and lower trehalose levels.

Feeding on susceptible or resistant rice can influence the transcriptome of BPH. Several DEGs have been reported in BPH fed on either susceptible or resistant rice by the comparative transcriptome method. For instance, by comparing the salivary gland transcriptomes of two BPH populations maintained on TN1 and Mudgo rice (resistant, contains *Bph1* gene), a total of 3757 unigenes were identified to exhibit differential transcript abundance [22]. Similarly, the comparative whole-body transcriptome of two BPH populations initially maintained on Mudgo and TN1 rice separately and fed on Mudgo rice identified 538 DEGs with important enriched functions, including sugar transport, development process, detoxification, insulin signaling, and cuticle formation [18]. In this study, the identified DEGs were functionally annotated with crucial physiological processes, such as sugar metabolism, sugar transport, detoxification, xenobiotic degradation, cuticle synthesis, and molting process (Table S3). For instance, feedings on the resistant PTB33 rice caused a significant upregulation of genes from the carboxylesterase family (*LOC111064061*, *LOC111048480*, *LOC111050119*, *LOC111058678*, *LOC111045597*, *LOC111047774*, and *LOC111045596*). Carboxylesterases play a vital role in metabolizing xenobiotics, including synthetic pesticides and the plant-produced toxic allelochemicals

in insect fat bodies and midguts [22,37–39]. Plant phloem sap can contain an array of allelochemicals to deter (antixenosis) or curb (antibiosis) the herbivory [22,40]. The PTB33 rice harboring three different BPH-resistance genes is known for its antibiosis gene function [3,24]. Thus, it is highly probable that the BPH could have been administrated while feeding on the PTB33 rice, and to rescue it from the xenobiotic toxicity, the carboxylesterase activities could have been increased. Additionally, on the other hand, genes with functions, such as calcium binding (*LOC111050001, LOC111060644, LOC111051627, LOC111047883, LOC111054785, LOC111056546*, and *LOC111054639*), were found to be significantly upregulated in the BPH fed on PTB33. Proteins with calcium-binding activity have been identified in BPH as effectors that can suppress rice resistance. The salivary EF-hand calcium-binding protein, NlSEF1, and the salivary calmodulin, NlCaM, with calcium-binding activity are two well-characterized effectors from BPH [28,41]. In our study, it is possible that some of these upregulated genes with calcium-binding activity might act as effector(s). Thus, possibly by inducing the expression of these genes, the BPH might be trying to suppress the PTB33 resistance. However, further in-depth research is required to establish this point. On the other hand, several genes annotated with cuticle synthesis and juvenile hormone binding protein (JHBP) were found to be downregulated in the BPH fed on PTB33 rice. Both these genes are associated with insect growth and metamorphosis, and their downregulation can lead to severe implications [42,43]. Additionally, JHBP has been reported to regulate the innate immune defenses in insects [44]. Thus, the PTB33 rice could be inducing the downregulation of these important BPH genes by using the allelochemicals or by any other means to hinder the normal growth, development, and even defense response in BPH (antibiosis). The resistant mechanism of the *Bph* genes present in PTB333, including *BPH2, BPH17-ptb*, and *BPH32*, are different. *BPH2* and *BPH17-ptb* confer resistance against BPH by antibiosis and antixenosis, whereas *BPH17-ptb* and *BPH32* contribute to tolerance. On the other hand, none of the reported *Bph* genes [*Bph13(t), BPH31, Bph6, Bph34Bph3, bph4, bph7*, and *BPH18(t)*] to confer resistance against BPH biotype 4 is mapped to PTB33 rice [45]. Thus, it could be possible that the resistance response of PTB33 rice against the BPH biotype 4 could be governed by some other pathways or by some unidentified genetic factors.

Finally, in our attempt to elucidate their respective functionality in BPH, we have silenced three genes independently via RNAi, including *NlCP1* (*LOC111046236*, involved in cuticle formation), *NlCYP320a1* (*LOC111060963*, involved in detoxification), and *NlTret1* (*LOC111058179*, involved in sugar transport). The silencing of *NlCP1* caused significant mortality in the BPH fed on TN1 or PTB33 rice as compared to the control group. Insect cuticle proteins serve as crucial structural proteins in insect development and physiology [42,46]. Therefore, the silencing of *NlCP1* might have hindered an array of physiological processes, including BPH locomotion, exoskeleton development, and resistance response. Similarly, the knockdown of *NlCYP320a1* encoding a cytochrome P450 monooxygenase brought down the BPH survival rates. The insect cytochrome P450 monooxygenase family plays an indispensable role in insect toxicology and resistance [47]. Thus, silencing the *NlCYP320a1* might have reduced the detoxification efficiency in BPH and caused mortality. Silencing another gene *NlTret1* encoding a trehalose transporter resulted in a significant reduction in the BPH survival rates. Trehalose is an important carbohydrate and crucial for BPH locomotion [31]. Lastly, silencing *NlTret1* could have hindered the transport of trehalose across the BPH body and might have altered the sugar metabolism and energy production affecting BPH fitness.

## 5. Conclusions

The current study reported the phenotypic, biochemical, and transcriptome dynamics of the Indian sub-continent biotype 4 BPH during a compatible and incompatible rice–BPH interaction. The overall BPH fitness level was challenged during the incompatible rice–BPH interaction as indicated by the amount of honeydew excreted and negative weight gain. Overall, biochemical assays confirmed that the nature of the interaction (compatible or incompatible) can influence the glucose and trehalose content and the sugar metabolism

in BPH. Furthermore, the comparative transcriptome analysis between the BPH fed on PTB33 rice or TN1 rice revealed several DEGs involved in crucial physiological processes, including sugar metabolisms, detoxification, cuticle development, molting, and xenobiotic metabolism. Furthermore, the RNAi-mediated silencing of the selected DEGs suggested their roles in BPH physiology and survival. Overall, the findings from this study will help to understand the complex rice–BPH interactions, and the data generated in this work will help in future studies, including potential effector identification in the BPH biotype 4.

**Supplementary Materials:** The following supporting information can be downloaded at: https://www.mdpi.com/article/10.3390/cimb45080429/s1, Table S1: List of primers used to evaluate the expression of the selected DEGs via RT-qPCR; Table S2: List of primers used to synthesize dsRNAs for the selected DEGs; Table S3: The list of all DEGs identified in the study. The significant upregulation is marked as "Up" while the down regulation is marked as "Down"; Table S4: The correlation table between the BPH_PTB33 and BPH_TN1 libraries as analyzed by the Spearman correlation analysis. Figure S1: (A) Rearing of BPH biotype 4 on TN1 rice; (B) Close up view of the BPH reared on TN1 rice; (C) Representative experimental set up for transcriptome sequencing. BPH are transferred on to PTB and TN1 rice separately and allowed to feed for 48 h; (D) Representative picture of the honeydew measurement experiment.

**Author Contributions:** Conceptualization, S.N.; validation, P.R. and S.N.; formal analysis, S.N., N.R. and D.G.; investigation, P.R. and S.N.; writing—original draft preparation, P.R., N.R. and S.N.; writing—review and editing, D.G. and S.N.; project administration, S.N.; funding acquisition, S.N. All authors have read and agreed to the published version of the manuscript.

**Funding:** This research was funded by the Science and Engineering Research Board (SERB), Government of India, with the grant number SRG/2021/000077.

**Institutional Review Board Statement:** Not applicable.

**Informed Consent Statement:** Not applicable.

**Data Availability Statement:** Data are contained within the article or Supplementary Materials.

**Acknowledgments:** P.R. acknowledges the financial assistance from SERB, India in form of the Project Associate-I fellowship. All authors express their gratitude to L.K. Rath for his help in obtaining different rice varieties. All authors express thanks to Centurion University for the necessary facilities and support.

**Conflicts of Interest:** The authors declare no conflict of interest. The funders had no role in the design of the study; in the collection, analyses, or interpretation of data; in the writing of the manuscript; or in the decision to publish the results.

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
