# Peer review of "Unveiling Nilaparvata lugens Stål Genes Defining Compatible and Incompatible Interactions with Rice through Transcriptome Analysis and Gene Silencing"

_cimb, doi:10.3390/cimb45080429_

Round 1
Reviewer 1 Report
Comments and Suggestions for Authors
The manuscript summarizes the results of a complex comparative study of the pattern of physiological and molecular characteristics of the Indian biotype 4 of brown planthopper (BHP) during feeding on the two rice varieties with contrast pattern of resistance to the pest. The authors identified a number of differentially expressed genes have been identified through a comparative transcriptome analysis of BHP in compatible and incompatible interactions, and in the experiments with RNA interference-mediated BHP methabolism and vitality.
It can be seen, that the manuscript is already carefully edited, the supplementary materials correspond to the content of the manuscript. To my opinion, the manuscript is ready for publication.
Notes:
Please include latin names of the planthopper and rice in the Abstract.
Please correct grammatical and stylistic errors on the lines 53-54, 126, 192-193.
The research aims at the characterization of fitness of the biotype 4 of brown planthopper (BPH), and at revealing genes the expression of which is differentially affected when the insect feeding on the host rice genotypes possessing and lacking pest resistance (Bph) genes.
I consider the topic is not highly original because the methodological approaches applied have been already used in the analogous studies on the biochemical and molecular characteristics during compatible and incompatible interaction between the host plants and other biotypes of the harmful organism, namely biotypes 1 and 2 which attack rice in China and East Asia. The object of the reviewed study was a highly virulent biotype 4 which causes severe damage of rice on the Indian subcontinent. Moreover, as the authors indicate, in the previous studies the salivary glands transcriptome was mainly studied whereas in the present research the wholebody transcriptome was analysed. Hence, the research addresses the specific gap in the understanding mechanisms of interaction of the very harmful biotype 4 BPH with a host rice plant.
The published material on the topic is very extensive but it concerns investigations of biotypes 1 and 2. The data for the biotype 4 are not so numerous. The phenotypic characteristics and transcriptome changes of the Indian sub-continent biotype 4 of BPH fed on two contrasting rice varieties, TN1 (susceptible) and PTB33 (resistant) were analyzed. It has been demonstrated in the present research that feeding on a resistant variety (incompatible interaction) significantly reduces BPH fitness. The results obtained are in corroborate with several previous studies that reported increased trehalose content in BPH while feeding on a resistant rice. In the reviewed study, the 1875 DEGs were identified and functionally annotated that is significantly higher than in a similar comparative study of other authors (Ji et al., 2013). Finally, in order to elucidate the functionality in BPH, the RNA interference-mediated independent silencing of three genes was obtained. The results of experiments can be considered as a proof of the importance of these genes for physiology and survival of BPH.
A question concerns the plant material selected for the study. The variety PTB333 possesses three genes of resistance to BPH (BPH2, BPH17-ptb, and BPH32). Resistance mechanisms conferred by these genes are different: BPH2 and BPH17-ptb determine antibiosis and antixenosis response whereas BPH17-ptb and BPH32 define tolerance (Cuong et al., 2021). According to literature sources, these genes confer resistance to the biotypes 1 or 2 of BPH. The different genes of resistance against biotypes 4 were reported: Bph13(t), BPH31, Bph6, Bph34Bph3, bph4, bph7, BPH18(t) (Muduli et al., 2021). Therefore, the results indicate that the variety PTB333 probably possesses new earlier nonidentified genetic factors conferring resistance to biotype 4 BPH. I consider this should be definitely pointed in the Discussion section or in the Conclusions.
The conclusions are generally consistent with the experimental data and arguments and address the main question.
All the references are appropriate.
The manuscript contains six figures and one table. All figures have good quality, the legends are clear and informative.
Additional notes: 1) Please include latin names of the planthopper and rice in the Abstract. 2) Please correct grammatical and stylistic errors on the lines 53-54, 126, 192-193.
Recommendation: 1)Accept after editing and taking into account my comments in the paragraph 4. 2) Taking into account not very high originality of the research which has been performed on the two (only) contrasting varieties I would recommend the Academic Editor to accept this manuscript as a short communication.
Author Response
Thank you for all the comments and suggestions. We have provided a point-to-point response to all the comments. "Please see the attachment."

Reviewer 2 Report
Comments and Suggestions for Authors
Manuscript entitled “Unveiling Nilaparvata lugens genes defining compatible and incompatible interactions with rice through transcriptome analysis and gene silencing” by Priyadarshini Rout et al. analyzes the transcriptomic, physiological, and gene silencing responses of the BPH during the compatible and incompatible rice-BPH interactions.
The work is interesting but, in my opinion, the manuscript presents some issues mainly in the Figures, Tables, Materials and Methods, and References, and, as well as other things, that need to be checked to make this work suitable for publication.
1. Main issues:
1.1. Regarding Figures.
In both tables and figures, the legends must be self-explanatory and contain all the necessary data.
In the legend of figures 1, 2, 3 (Rt-qPCR), 6 and 7 the “n” are missing.
Figure 3. The different LOC references are not known to which points they correspond. what are the triangles? There are no grey points (or are they dark grey?), what are the green and blue points?
In the ZIP file there is a figure S1 which is not listed in the text and which is the same as figure 5.
1.2. Regarding Tables.
Figures and Tables should be number according to its citation in the main text.
Table S2 (in line 138) is the Table S1.
Table S3 (in line 157) is the Table S2.
Table S1 (in line 216, line 231, line 353) is the Table S3
Please, change the order in lines 413 – 415.
1.3. Regarding section Materials and Methods
I must point to the authors that, according to the Instructions for Authors section of this journal, Full experimental details must be provided so that the results can be reproduced. The Materials and Methods section of this manuscript is not detailed enough for other authors to repeat some of the experiments, in particular the construction of the cDNA libraries (line 119).
1.4. Regarding section References
The references list has numerous format mistakes. According to the “Instructions for Authors” (Journal Articles - Author 1, A.B.; Author 2, C.D. Title of the article. Abbreviated Journal Name Year, Volume, page range.) the references list has numerous format mistakes.
The correct form of Abbreviated Journal Name is for example in reference 1 “Int. J. Mol. Sci.” and not “Int J Mol Sci”.
Please revise all the text and correct the missing spaces (e.g. reference 1 “Sci2018”, Reference 3 “Basel)2020”
The volume should be in italics.
Revise the italics (e.g. Line 450 Pleas correct “Nilaparvata lugens (Stål)” with “Nilaparvata lugens (Stål)”, revise the entire manuscript).
The DOI of each reference must be removed.
Almost all the words in the title begin with capital letters. Many of them should not be in that format (reference 1, 2, 3, 4, etc.). Please revise all references (reference 15 is correct for example).
The reference 2 is a Book Chapters and the format is wrong. According to the “Instructions for Authors” Books and Book Chapters (Author 1, A.; Author 2, B. Book Title, 3rd ed.; Publisher: Publisher Location, Country, Year; pp. 154–196.
or
Author 1, A.; Author 2, B. Title of the chapter. In Book Title, 2nd ed.; Editor 1, A., Editor 2, B., Eds.; Publisher: Publisher Location, Country, Year; Volume 3, pp. 154–196.
Etc.
Please revise all the references
2. Additional comments
Line 2. Please correct “Nilaparvata lugens genes” with “Nilaparvata lugens Stål genes”
Line 5. Please correct “1*” with “1,*”
Lines 26 and 27. Please correct “Rice, Brown planthopper, Transcriptome, Compatible and incompatible interaction, 26 RNAi” with “rice; brown planthopper; transcriptome; compatible and incompatible interaction; 26 RNAi”.
Line 92. Please correct “ICAR-National” with “Indian Council of Agricultural Research-National”.
Line 96. Please correct “2 ºC” with “2ºC”
Line 115. Please check “oC.” Is not similar to ºC (in line 96).
Line 118. Please correct “0.8 %” with “0.8%”
Line 137 Please correct “rps15genes” with “rps15 genes”
Line 145. Please correct “(PBS, 0.02 M, pH 6.0).” with “(PBS 0.02 M, pH 6.0).”
Line 146. Please correct “3000” with “3,000”
Line 146. Please correct “4 ºC” with “4ºC”
Line 149. Please correct “13000” with “13,000”
Line 149. Please correct “4 ºC” with “4ºC”
Line 178 and all the text and leyends. Please correct “P< 0.” with “P < 0.”
Line 215. Please correct “1875” with “1,875”
Line 224. Please correct “Padj< 0.” with “Padj < 0.”
Line 291. Please correct “4DPI” with “4 DPI”
Line 346. Please correct “3757” with “3,757”
Lines 417 to 419. Please correct the abbreviations of the authors. E.g. “S.N.” and not “SN”.
Author Response

(The authors gave the same response as above.)

Reviewer 3 Report
Comments and Suggestions for Authors
The article "Unveiling Nilaparvata lugens genes defining compatible and incompatible interactions with rice through transcriptome analysis and gene silencing" by Priyadarshini Rout, Nihal R, Dinkar Gaikwad and Satyabrata Nanda provides evidence to support the role of gene silencing in plant resistance to insect damage.
This topic is very relevant, especially given the important object for nutrition - rice.
Unfortunately, in the introduction and discussion, little attention is paid to the fundamental aspect that ensures this process of epigenetic regulation of gene expression. I think the authors need to strengthen this aspect especially in light of methylation features.
I also think that the presentation of materials on the provision of the experiment (photo) and the features of the "work" of the pest would be important for this article.
I also recommend improving the quality of figure 5, since its resolution is not enough.
A correlation table of the obtained data would also be useful.
Author Response

(The authors gave the same response as above.)
